# Phase-Separated Subcellular Compartmentation and Related Human Diseases

**DOI:** 10.3390/ijms23105491

**Published:** 2022-05-14

**Authors:** Lin Zhang, Shubo Wang, Wenmeng Wang, Jinming Shi, Daniel B. Stovall, Dangdang Li, Guangchao Sui

**Affiliations:** 1College of Life Science, Northeast Forestry University, Harbin 150040, China; linzhang@nefu.edu.cn (L.Z.); shubowang@nefu.edu.cn (S.W.); wangwenmeng@nefu.edu.cn (W.W.); jmshi@nefu.edu.cn (J.S.); 2College of Arts and Sciences, Winthrop University, Rock Hill, SC 29733, USA; stovalld@winthrop.edu

**Keywords:** liquid–liquid phase separation (LLPS), membraneless organelles, phase-separated condensates, human diseases

## Abstract

In live cells, proteins and nucleic acids can associate together through multivalent interactions, and form relatively isolated phases that undertake designated biological functions and activities. In the past decade, liquid–liquid phase separation (LLPS) has gradually been recognized as a general mechanism for the intracellular organization of biomolecules. LLPS regulates the assembly and composition of dozens of membraneless organelles and condensates in cells. Due to the altered physiological conditions or genetic mutations, phase-separated condensates may undergo aberrant formation, maturation or gelation that contributes to the onset and progression of various diseases, including neurodegenerative disorders and cancers. In this review, we summarize the properties of different membraneless organelles and condensates, and discuss multiple phase separation-regulated biological processes. Based on the dysregulation and mutations of several key regulatory proteins and signaling pathways, we also exemplify how aberrantly regulated LLPS may contribute to human diseases.

## 1. Introduction

To organize complex biochemical reactions in a cellular environment, cells create compartments, or organelles. A compartment needs a boundary to separate it from the surroundings, and the components within it are mostly able to freely diffuse, so that biological processes can take place inside [1]. Many compartments, such as the endoplasmic reticulum and Golgi apparatus, are organelles surrounded by lipid bilayer membranes. However, many other cellular compartments are not restricted by any membrane, such as nucleoli, Cajal bodies, PML nuclear bodies, stress granules and germ granules [2,3,4,5,6]. In a cell, these compartments harbor a variety of biomolecules with specific functions in a spatiotemporally controlled manner to ensure undisturbed biological processes and fulfill designated cellular functions [7]. In the past decade, accumulating studies suggest a physical process, known as phase separation, that can drive the assembly of these membraneless compartments. The concept that liquid-–liquid phase separation (LLPS) may be generally involved in many cellular processes has been gradually uncovered and increasingly appreciated.

Phase separation is a common phenomenon in physics and chemistry: two liquids do not compatibly dissolve in a homogeneous liquid phase, resulting in a distinct phase–phase separation state. In other words, a uniformly mixed and supersaturated solution without further dispersion will spontaneously separate into a dense phase and a dilute phase that can stably coexist. The droplets or condensates produced by LLPS are different from ordinary droplets. For example, droplets composed of proteins and RNAs are not completely uniform, such as nucleoli with three layers regulating different stages of ribosomal biogenesis, but show the characteristics of liquid flow [8]. LLPS is quickly accepted as a key and general mechanism underlying the creation of biomolecular condensates that can promote the formation of membraneless organelles to regulate various cellular functions and activities [9]. However, phase separation is highly sensitive to altered physical and chemical conditions. For example, many protein condensates are regulated by environmental factors that determine the strength and valency of intermolecular interactions, including temperature, pH, salt concentration, component concentration and composition [10]. A molecule may need to reach a threshold concentration to initiate LLPS, and even a small difference in temperature and protein, nucleic acid or salt concentration can lead to distinct outcomes [11]. Moreover, the presence of crowding molecules, such as polyethylene glycol (PEG), dextran and ficoll, can greatly enhance the process of LLPS [12]. In compositional studies of different membraneless organelles, proteins and nucleic acids may utilize multivalent interactions to form phase-separated condensates with designated physical and chemical properties different from the originally uniform cellular environment. Many key regulatory proteins have been reported to undergo phase separation, of which the dysregulation has been etiologically associated with the onset and progression of many diseases, such as amyotrophic lateral sclerosis (ALS), Alzheimer’s disease, Huntington’s disease and different cancers [13,14]. In the current review, we summarize recent studies of phase-separation-mediated compartmentation, and discuss how aberrantly regulated LLPS causes human diseases, especially neurodegenerative disorders and cancers.

## 2. Biomolecular Condensates

Biomolecular condensates are commonly present in live cells, and they troubled scientists for many decades as they attempted to elucidate their formation and functions. Phase separation provides a mechanism for the formation of these condensates that separate or isolate different molecules with related activities in defined compartments. It has also been proposed that the ability to undergo LLPS may be a universal property of proteins and nucleic acids under specific circumstances [15,16,17].

### 2.1. The Molecular Features of Biomolecular Aggregates

Many studies indicate that phase separation requires the establishment of an interactive network through multivalent protein molecules that are composed of multiple modular interactive domains and/or contain disordered regions [18]. The interactions include charge–charge, cation–π, π–π stacking and hydrogen bonds, involving both side chains and backbones of the proteins. For example, Nephrin, Nck and Neural Wiskott–Aldrich syndrome protein (N-WASP) can be assembled into a highly ordered and multivalent protein complex through the interactions between phosphorylated tyrosines of Nephrin and SH2 domains of Nck, and between SH3 domains of Nck and proline-rich motifs of N-WASP [19,20,21].

The phase separation phenomenon has unique physical characteristics, including fluidity, fusion and fluorescence recovery after photobleaching when fused with a fluorescent protein. Meanwhile, the formation of droplets is generally both concentration- and valence-dependent. Intrinsically disordered regions (IDRs) are featured characteristics of many proteins with LLPS capability, and are often both necessary and sufficient for the formation of phase-separated condensates. IDRs usually have low complexity and contain homo-polymeric repeats of specific amino acids, such as glycine, serine, proline and glutamine, with strong self-sustaining aggregation potentials [22,23]. Recently, we reported that histidine clusters could decide the phase separation of several proteins, including YY1, HOXA1, FOXG1B, ZIC3 and HNF6 [24]. Several algorithms have been developed to help researchers predict IDRs in a protein [25,26]. However, not all highly scored sequences based on the prediction software could necessarily form phase-separated condensates [27]. Meanwhile, IDR mutations are causally related to various human diseases, such as cardiovascular disorders, cancers and neurodegenerative diseases [27,28]. Vacic et al. investigated about 100,000 annotated missense disease mutations and discovered that 21.7% of them were located in the IDRs [29]. Among these mutations, 20% led to disorder-to-order transitions, such as increased α-helical propensity, significantly higher than those of annotated polymorphisms and neutral evolutionary substitutions [29].

A classic example is the correlation between fused in sarcoma (FUS) mutations and neurodegenerative diseases, including ALS, essential tremor and rare forms of frontotemporal lobar degeneration [30]. FUS protein contains a prion-like domain that is intrinsically disordered and can form liquid compartments in both the nucleus and cytoplasm [31]. Multiple FUS mutants exhibit significantly reduced mobility and eventually cause prion-like propagation of proteinaceous aggregates in neurons and glial support cells, characteristic of ALS [32]. Another example is the MutL Homolog 1 (MLH1) protein that is essential in DNA mismatch repair. The residue V384 located in the disordered segment of MLH1 is the most common site of mutations. The mutant MLH1 (V384D) is associated with increased susceptibility to colorectal cancer and is prevalent in HER2-positive luminal B breast cancer [33,34]. Phase separation is also involved in the antiviral immune response against the novel severe acute respiratory syndrome (SARS) coronavirus 2. The nucleocapsid protein of SARS2 may undergo LLPS with RNA and subsequently reduced Lys63-linked polyubiquitination and aggregation of mitochondrial antiviral-signaling protein (MAVS), which suppresses the innate antiviral immune response [35].

### 2.2. Materials Properties of Phase-Separated Condensates

LLPS contributes to the assembly of different membraneless organelles with different functional commitments in cells [36]. Whether a macromolecule can undergo phase separation depends on its concentration and property, as well as environmental conditions, such as pH, temperature, salt type and concentration. Meanwhile, phase-separated condensates formed under a particular physiological circumstance are accessible to various, but also selective, molecules in cells. The condensation process through the LLPS mechanism is generally reversible with a mobile liquid-like dense phase, and constant exchanges between the dense and light phases. However, the phase-separated condensates are subject to further transitions, such as gelation to form hydrogel that is virtually irreversible under physiological conditions. Whether LLPS condensates remain in a liquid and mobile state or become gelatinous and even solidified are physiologically or pathologically relevant [1,22,37,38]. We have illustrated previously reported membraneless organelles with their subcellular localization and functions in Figure 1. Meanwhile, we also summarized their sizes, components, functions and related diseases in Table 1. Here, we discuss the formation, compositions and other properties of several membraneless organelles and key regulatory protein-mediated condensates in the context of human diseases.

#### 2.2.1. Stress Granules

Both stress granules and processing bodies (P bodies) are composed of RNA and protein molecules that drive the phase separation of these membraneless organelles. Stress granule formation is exclusively induced by stress signals imposed on the cells, while P bodies can be constitutively visible in many cell types, but their size and number may increase in response to stress [40]. Stress granules contain translation-initiation molecules, and P bodies harbor factors regulating mRNA degradation, but they share many common proteins related to RNA metabolism. Mechanistically, in response to certain stresses, translation initiation can be stalled and ribosomes will disassociate from mRNA, which is the so-called ribosome run-off phenomenon. The released mRNA binds to RNA-binding proteins (RBPs) that promote stress granule formation. The mechanism of constitutive presence of P bodies remains unclear, but the stress-induced retardation of translation preinitiation directly contributes to their increased size and number [40].

Dysregulation of stress granules and P bodies is causally related to different diseases. Stress granules are considered an adaptive response of cells to acute stress, and their formation, composition and life span are associated with cancers, heart diseases, neurodegenerative disorders, inflammatory diseases and viral infections [66]. The oncogenic process consists of hypoxia, ER stress and osmotic alterations that all constitute the signals to induce stress granule formation. Meanwhile, chemotherapeutic challenges can also induce the assembly of stress granules, which contributes to the development of chemoresistance and metastasis of cancer cells [66]. Thus, drugs, such as 15d-PGJ2 targeting the eukaryotic initiation factor 4A-I (eIF4A1) in the stress granules, can inhibit proliferation and induce apoptosis of leukemic and colorectal cancer cells [67]. Several neurodegenerative diseases are caused by dysregulated stress granules that generally exhibit increased formation or reduced dissociation of stress granules compared to in the cognate normal cells. In particular, genetic mutations of certain RNA-binding proteins may impair stress granule assembly and composition leading to neurodegenerative diseases. For example, an FMRP mutant with defective stress granule assembly represents an etiologic cause of the Fragile X syndrome with mild-to-moderate intellectual disability [68]. Mutations in other stress granule-associated RNA-binding proteins are also discovered in Alzheimer’s disease patients [69]. In addition, the neurons of Alzheimer’s disease patients exhibited pathological aggregates by the nucleation of the proteins in stress granules, such as TIA1/R and G3BP1 [69]. During viral infection, many viruses can use a special viral protease to cleave essential stress granule proteins, which can circumvent the cellular defense against viral infection [70,71].

#### 2.2.2. P Bodies

As another type of cytoplasmic ribonucleoprotein granules, P bodies are relatively understudied for their relevance to human diseases, although current evidence strongly suggests their involvement in neurodegenerative disorders, viral infection and autoimmune diseases. Mutations of DDX6 disrupt P body assembly, which is causally linked to intellectual developmental disorders with impaired language and dysmorphic facies [72]. In response to infection by RNA viruses, the number and stability of P bodies may change, and their components may be recruited to viral replication centers, although the underlying mechanisms remain unclear [73]. In addition, autoantibodies against P body components have been reported to contribute to autoimmune diseases [74,75].

#### 2.2.3. Nucleolus

The nucleolus is an important membraneless organelle consisting of ribonucleoproteins and RNAs is assembled in multilayers through the LLPS mechanism [55]. In the past century, the roles of the nucleolus in hosting RNA polymerase I-mediated transcription, ribosomal RNA (rRNA) modification and processing, and rRNA complex assembly have been gradually recognized. A nucleolus of a mammalian cell may contain several functional modules, each of which constitutes three subcompartments or layers. From the inner to periphery, the three layers include the fibrillar center, the dense fibrillar component and the granular component, responsible for different steps of ribosomal biogenesis [55]. The nucleolus is separated from other compartments of the nucleus; however, due to the membraneless status, the nucleolus harbors various contents that dynamically exchange with the remaining nuclear components. Therefore, nucleoli are important organelles for transient sequestration of crucial factors involved in various biological functions, including the responses to genotoxic and oxidative stress, heat shock, starvation, oncogenic insults and viral infection [55,76]. These stresses may affect the shape, size and number of nucleoli, and the diseased states can markedly alter nucleolar morphology. Interestingly, despite the relatively isolated compartment and spatially distinct layers of each nucleolus, spontaneous coalescence may occur when two nucleoli have intimate contact, resembling droplet fusion during LLPS. Meanwhile, many nucleolar proteins contain IDRs, which are especially enriched by positively charged arginine and lysine residues [77,78].

Dysregulation of the nucleolus may aberrantly change nucleolar morphology, size and number per nucleus, and is tightly linked to various diseases. Excessive production of ribosomes by nucleoli may drive oncogenic transformation. On the other hand, defective activity of ribosome biogenesis may cause a shortage of properly formed ribosomes, and even cause aberrant nucleolar hardening, leading to reduced rRNA and ribonucleoprotein processing. These kinds of ribosomopathies may eventually cause different diseases, such as muscle atrophy and X-linked subtype of dyskeratosis congenita [79,80]. A hexanucleotide repeat GGGGCC (or G4C2) is present in an intron of the *C9ORF72* in chromosome 9, and its expansion can reach up to thousands of copies in ALS patients. Mechanistically, the expanded G4C2 sequence can generate arginine-containing toxic dipeptide repeats that promiscuously interact with the IDRs of RNA-binding proteins to form protein aggregates, and thus impair the dynamics of membraneless organelles, such us nucleoli, leading to the diseases [81]. In addition, the material state of the nucleolus is relevant to aging or longevity. Studies using *C. elegans* as a model revealed that both reduced rRNA production and knockdown of fibrillarin were associated with smaller nucleolar size and extended life span of the worm [82].

#### 2.2.4. Examples of Regulatory Proteins with LLPS Potential

Besides the reported membraneless organelles, many intrinsically disordered proteins, especially those with nucleic acid binding affinity, can form isolated compartments through the LLPS mechanism, and their dysregulation may undergo liquid-to-solid transitions, leading to various diseases [19,83].

The prion-like domains (PrLDs) have relatively low complexity, and are enriched in glycine and uncharged polar amino acids [84]. The PrLDs have been identified in about 240 human proteins, especially many RNA-binding proteins, such as FUS, EWSR1, TDF-43 and TAF15 that are etiologically related to several neurodegenerative diseases, including frontotemporal dementia and ALS.

The RNA-binding protein FUS has 526 amino acids and belongs to the FET (FUS, EWSR1 and TAF15) family. *FUS* was originally discovered to fuse with the *CHOP* gene, and the fusion oncoprotein promotes the development of round cell liposarcoma and myeloid leukemia [85]. In addition to an RNA-binding motif, FUS contains a highly conserved C-terminal nuclear localization signal (NLS) that may harbor various mutations discovered in patients [86]. The EWSR1 protein has a transcriptional activation domain at the N-terminus, and regulates gene expression, cell signaling, RNA processing and RNA transport. The chromosomal translocation between the *EWSR1* and *FLI* genes can produce an oncogenic fusion gene that accounts for about 90% of Ewing sarcomas [87].

Since the N-terminus of FUS contains the IDR, the FUS-CHOP fusion created more intensified nuclear puncta than FUS and CHOP alone, with incorporation of BRD4, a bona fide marker of super-enhancers. Similarly, LLPS is considered as a driving force for the *EWSR1-FLI* fusion gene to regulate transcription and initiate cell transformation [88].

### 2.3. Regulation of Condensate Assembly

The assembly and biophysical properties of LLPS condensates are precisely regulated by chaperone proteins, enzymes for post-translational modifications (PTMs) and other cellular factors [89].

#### 2.3.1. Effects of PTMs on Protein Phase Separation

Different PTMs, such as phosphorylation, acetylation, arginine methylation and SUMOylation that regulate protein–protein or protein–nucleic acid interaction strengths, are well-recognized key regulatory factors of phase separation. Furthermore, PTMs are engaged in the assembly and disassembly of condensates, as well as the regulation of their material properties. As a rapid and reversible process, phosphorylation is one of the most well-characterized PTMs modulating biomolecular phase transitions [90,91]. For example, in Alzheimer’s disease, phosphorylation of Tau, a microtubule-associated protein, alters the charge distribution to promote its electrostatic interactions, leading to the formation of Tau aggregates [92]. Additionally, phosphorylation hinders tubulin assembly within Tau condensates. Previous studies indicated that neuronal loss and memory impairment were causally related to the presence of highly phosphorylated soluble Tau protein [93].

Phosphorylation of α-synuclein (α-syn) at Tyr39 (pY39) is enriched in patients with Parkinson’s disease, and plays an important role in regulating the liquid–solid phase transition of α-syn [94]. pY39 can accelerate α-syn aggregation and inhibit its degradation through autophagy and proteasome pathways in cortical neurons. In general, α-syn phosphorylation may alter its fibril structure and exacerbate pathogenesis of Parkinson’s disease [94,95]. As discussed above, FUS is a protein tightly related to neuronal degeneration diseases. FUS phosphorylation at its IDR could disrupt its phase separation and cytoplasmic aggregation, which reduces FUS-associated cytotoxicity [96], suggesting that FUS is a potential therapeutic target in the treatment of neurodegenerative diseases. In addition, the interactions between tyrosines in the IDR and arginines in the C-terminal regions of the FUS protein are crucial to its phase separation. The methylation of these arginines disrupts these interactions, leading to reduced FUS phase separation; however, hypomethylation of these arginines strongly promotes FUS phase separation and gelation, leading to the formation of immobile hydrogels stabilized by intermolecular β-sheets. The loss of FUS mobility causes impairment of neuron terminals and leads to the disease manifestation of frontotemporal lobar degeneration [97].

Polycomb repressive complexes (PRCs) are important regulators for gene repression during embryonic development and oncogenic progression [98]. In *C. elegans*, a polycomb protein SOP-2 functions as the counterpart of the human PRC1 complex to regulate *HOX* gene expression [99]. Qu et al. reported that SOP-2 contained an IDR and could form phase-separated droplets. Importantly, sumoylation at K453 and K594 SOP-2 could allow it to produce droplets with increased sizes and abundancy, and slightly improved internal mobility compared to the droplets formed by the unmodified protein [100]. Sumo-conjugation is likely essential for both phase separation and transcriptional regulation of SOP-2, because its sumoylation is required for both its localization into nuclear bodies and physiological repression of the *HOX* genes [101].

Phase separation-mediated formation of membraneless organelles is cell-cycle-dependent. Most membraneless organelles are dissolved when the nuclear envelope breaks down during mitosis, but are reformed as mitosis is completed. The kinase activity of DYRK3 plays an important role in dissolving several types of membraneless organelles during mitosis [102]. In fact, DYRK3 has been demonstrated to cause the dissolution of stress granules upon stress relief [103], and this activity is dependent on DYRK3’s association with HSP90. In the absence of the heat-shock protein, the inactive DYRK3 either stays in stress granules or undergoes degradation [104].

#### 2.3.2. Effects of Chaperones on Protein Phase Separation

Molecular chaperones play a key role in the assembly of phase-separated condensates. The historically recognized functions of chaperones are their abilities to promote correct protein folding and subsequently prevent protein aggregation into nonfunctional structures. A number of recent studies have revealed the activity of molecular chaperones, including several heat shock proteins, to regulate phase separation [105].

Chaperones regulate protein–protein interplay and assist in protein folding through directly interacting with them in an energy-consuming manner [97,106]. Molecular chaperones, including many heat shock proteins, are extensively involved in the maintenance of intracellular protein homeostasis. Previous studies indicate the presence of different heat shock proteins in a variety of membraneless organelles, such as HSP40, HSP70, HSP90, etc. Gu et al. reported that classes I and II of the HSP40 proteins could undergo phase separation due to their contents of flexible regions enriched with glycine and tyrosine [107]. DNAJB1, a member of the class II HSP40 proteins, could form condensates in nuclear bodies. In response to stress, DNAJB1 can translocate into stress granules. Interestingly, when cophase-separated with FUS, DNAJB1 can prevent FUS from forming amyloid fibrils in vitro and reduce aberrant FUS aggregation in cells [107]. As discussed above, hypomethylation of arginines in the C-terminus of FUS facilitates its phase separation and gelation. However, transportin 1 can serve as a chaperone protein of FUS to reduce its granule formation without affecting its methylation status, and eventually rescue attenuated protein synthesis caused by FUS aggregation in axon terminals [97].

As a canonical small chaperone, HSP27 localizes in stress granules. Due to the interaction with the IDR of FUS, HSP27 can reduce its LLPS. In addition, stress can induce HSP27 phosphorylation that subsequently promotes its co-phase separation with FUS. The presence of HSP27 can prevent FUS from forming amyloid fibrillar aggregates, and thus preserve its liquid phase [106]. Consistently, when mice of an Alzheimer’s disease model were crossed with human *HSP27* transgenic mice, overexpressed HSP27 could rescue multiple neurodegenerative defects of the disease, including impaired spatial learning, increased neuronal excitability, reduced long-term potentiation, and widespread amyloid deposition in the brains [108].

As a histone chaperone, CAF-1 has LLPS properties and can form nuclear bodies through recruiting histone modifiers and other chaperones, which contributes to the establishment and maintenance of HIV-1 latency. Therefore, disruption of phase-separated nuclear bodies of CAF-1 can potentially reactivate latent HIV-1 to eradicate the viral reservoir caused by its latency [109].

### 2.4. Functions of Phase-Separation Condensates

LLPS have been reported to be involved in various biological processes and regulations. We summarize the LLPS-associated functions into the following four categories.

#### 2.4.1. Regulation of Biological Reactions

In cells, the coordinated processes of biochemical reactions benefit from both membrane-restricted and membraneless organelles. The membraneless particles or condensates formed by LLPS are rich in selective proteins and nucleic acids, increasing their local concentrations and subsequently accelerating biochemical reactions [38].

Strulson et al. mimicked the intracellular compartmentalization by partitioning RNA in an aqueous two-phase system established by PEG and dextran. The RNA molecules could show up to 3000-fold enrichment in the dextran-rich phase, and compartmentalization could enhance the rate of ribozyme cleavage by 70-fold [110]. The histone locus body (HLB) is an evolutionarily conserved nuclear body with enriched protein and RNA factors required for histone gene transcription and pre-mRNA processing [111]. In this liquid-like compartment, many factors, such as FLASH and U7 snRNP, essential and constitutive components in HLB, exhibit greatly increased concentrations over the levels in the exterior cellular environment [112].

MicroRNAs (miRNAs) can promote mRNA degradation and/or block translation through targeting the 3′-UTRs. In this regulation, the formation of a miRNA-induced silencing complex (miRISC) consisting of multiple proteins is crucial to the miRNA-mediated gene repression. AGO2 and TNRC6B are the core components of the miRISC. The glycine/tryptophan (GW)-rich domain of TNRC6B is an intrinsically disordered region that promotes phase separation through multivalent interactions with three tryptophan-binding pockets in the PIWI domain of AGO2 [113]. The phase-separation process can enrich both AGO2 and TNRC6B in the condensates, and sequester RNAs to be degraded, which accelerates AGO2-mediated deadenylation of target RNAs.

In addition to the compartmentalizing phenomena discussed above, many other LLPS-mediated membraneless organelles, such as Cajal bodies, nucleoli and PML bodies, can concentrate proteins and nucleic acids involved in different designated biological processes in a confined space, which can enhance both reaction rates and efficiency [114].

The LLPS may also provide a platform that allows nascent proteins to quickly associate with their functional partners, which may determine their activities and destinies. Ma et al. reported the membraneless TIS granules formed by an RNA-binding protein TIS11B, which could partially cover of the cytoplasmic side of the rough endoplasmic reticulum (ER) [53]. The integration of these TIS granules and the ER can generate subcellular compartments, termed as TIS granule-ER, or TIGER, that constructs a biophysically and biochemically distinct environment from the cytoplasm. The TIS granules can promote the association between the SET protein and membrane proteins to be translated, such as CD47 and PD-L1, through a mechanism that the 3′-UTRs of the mRNAs of the membrane proteins facilitate the interaction between SET and CD47 or PD-L1. As a result of the SET-binding, the cell surface expression of the CD47 or PD-L1 can be significantly enhanced, which determines the cell identity. This discovery revealed an exciting notion that protein functions can be regulated by the lengths of the 3′-UTRs. In other words, proteins with the same amino acid sequence but encoded by mRNA isoforms with alternative 3′-UTR lengths may have different functions or subcellular localizations [115]. Therefore, 3′-UTRs may act as a medium or scaffold to nurture nascent proteins, and qualitatively change their properties and fates. Noteworthily, it has been reported that over 50% of protein-coding genes can generate mRNA isoforms with alternative 3′-UTRs [116]. Whether the nurturing niche provided by the TIS granules or TIGER compartments can be generalized to the regulation of the functions, localizations or fates of other proteins, in addition to CD47 and PD-L1, is a very intriguing question and deserves future exploration.

#### 2.4.2. Regulation of Gene Expression

RNA polymerase II (Pol II) is responsible for the transcription of mRNAs and many noncoding RNAs, such as lncRNA and microRNAs. RNA Pol II has a highly conserved C-terminal domain (CTD) that contains 52 repeats of the YSPTSPS heptapeptide essential to polymerase activity [117]. The hyperphosphorylation of the CTD mediated by CDK9 can stimulate target gene transcription. As the kinase component of the positive transcription elongation factor b (P-TEFb), CDK9 can release the paused Pol II at a promoter periphery and facilitate its entry to the gene body, to achieve transcriptional elongation. CDK9 also regulates transcription termination through phosphorylating a Pol II-associated protein, SPT5, and promoting its interaction with the poly(A) site [118,119].

Lu et al. reported that a phase-separation mechanism is also critical for CTD hyperphosphorylation that activates RNA Pol II [117]. Despite the inclusion of a low-complexity region, the isolated CTD of RNA Pol II does not undergo phase separation by itself. However, the CTD can be trapped by the phase-separated condensates formed by the IDR of cyclin T1 that interacts with CDK7. Through this interaction, cyclin T1 compartmentalizes CKD7 and the CTD in restricted condensates to facilitate the hyperphosphorylation reaction of RNA Pol II. Additionally, the CTD can also bind to the low-complexity domains of transactivating proteins FUS, TAF15 and hnRNPA2 to form nuclear granules that promote transcription [120,121].

In the past few years, a rapid surge of studies has demonstrated that many transcription factors and coactivators are able to undergo phase separation that can help them create dynamic hubs, clusters or condensates to regulate target gene expression (Figure 2). Some of these condensates can be assembled into super-enhancers with many tandemly adjacent enhancers, each of which is typically 50 to 1500 base pairs in length [122]. The transcription factors, such as OCT4 and GCN4, harbor IDRs in their transactivation domains that can undergo phase separation to form clustered enhancers or super-enhancers and activate gene expression [122]. Meanwhile, many coactivators, such as BRD4, MED1 and p300, act as key components of the enhancer complexes that drive the expression of the master genes to determine cell identity or promote oncogenesis [11]. As we recently reported, the transcription factor YY1 has an IDR featured with an 11-histidine cluster. Deletion of the histidine cluster or replacing it with 11 alanines abolishes YY1’s ability to form nuclear puncta and even deprive its dominant nuclear localization. Through the phase-separation mechanism, YY1 compartmentalizes many coactivators, including p300, BRD4, MED1 and CDK9, to assemble clustered enhancers that activate *FOXM1* gene expression and contribute to mammary tumor formation in a mouse model [24].

Another example is the coactivator YAP that can cause chromatin reorganization to activate its target genes. In this regulation, YAP forms phase-separated condensates to compartmentalize the transcription factor TEAD1 and other coactivators, such as TAZ. The YAP condensates in the nucleus consist of super-enhancers with an accessible chromatin structure [123].

Signal transducer and activator of transcription 3 (STAT3) is a transcription factor regulating the expression of a variety of genes involved in different biological processes. As a key regulator in the anti-cancer immune response, STAT3 can be activated by various cytokines. Aberrant activation of STAT3 has been observed in many cancers, which serves as a bona fide target in cancer therapies [124]. Early studies indicated that tyrosine phosphorylation of STAT3 stimulated by interleukin 6 could cause its translocation into nucleus where STAT3 was activated, bound to the enhancer elements of target genes, and formed nuclear bodies. Thus, it was proposed that the STAT3 nuclear bodies could either be directly involved in activated gene transcription or serve as reservoirs of activated STAT3 [125]. Recent studies revealed that the biomolecular condensates formed by activated STAT3 exhibited LLPS properties, suggesting that the phase-separation mechanism contributes to STAT3-mediated gene activation [51,126].

#### 2.4.3. Regulation of Viral Infection

Many studies have demonstrated the regulatory roles of LLPS in both the viral life cycle and virus–host interactions [17,127]. Viral proteins with IDRs can promote the formation of membraneless compartments used for the replication of viruses. These compartments are enriched with specific proteins and nucleic acids, and serve as “viral factories” for the replication, assembly and trafficking of viruses. The LLPS condensates are selective for the inclusion or exclusion of components to allow optimal viral production, and may also avoid the defense of the host immune system. For example, cells infected by negative-strand RNA viruses, such as rabies virus (RABV), rotavirus, vesicular stomatitis virus (VSV), Ebola virus, measles virus, influenza A virus and respiratory syncytial virus (RSV), may form cytoplasmic LLPS condensates that allow all the ribonucleoparticle (RNP) components and viral RNAs to be synthesized inside and assembled into viral particles [128,129,130,131,132,133,134]. A report by Fouquet et al. revealed that the phosphoprotein P, essential for viral transcription and replication of RABV, could shuttle between the cytosol and the Negri bodies formed by the virus, leading to the recruitment of focal adhesion kinase (FAK) and HSP70, two cellular proteins with proviral activities [135].

Viral protein-mediated LLPS can interfere with the functions of host cells through two mechanisms, either regulating the expression of cellular genes or modulating the activities of cellular proteins. The oncogenic effects of Epstein–Barr virus (EBV) can be used as an example of the first mechanism. EBV is a human virus with potent activities to induce malignant transformation of infected cells through the activation of both viral oncogenes and cellular proto-oncogenes [136,137]. EBNA2 and EBNALP are two EBV-encoded transcription factors that form nuclear puncta using their IDRs, leading to the formation of super-enhancers on the promoters of the oncogenes MYC and RUNX3 to promote their transcription and subsequent oncogenesis [138]. In contrast, the functional interplays between viral and cellular proteins in the context of LLPS have been relatively understudied [17,127]. The formation of fibrillar aggregates by viral proteins may exert various effects on host cells, including inhibition of key cellular processes, such as such as necroptosis, and sequestration cellular transcription factors to block host cell RNA synthesis [17,139].

LLPS-related mechanisms not only mediate the impairments of infected cells caused by viruses, but also contribute to the defense system of host cells against viral infection. Human myxovirus-resistance protein A (MxA) is a cytoplasmic dynamin-family large GTPase with a molecular weight of about 70 kDa, and can be induced by 50- to 100-fold when cells are treated by type I and III interferons [140]. MxA associates with the endoplasmic reticulum and Golgi apparatus, and exhibits antiviral activity against several RNA and DNA viruses. A study by Davis et al. demonstrated that MxA formed metastable membraneless cytoplasmic spherical or irregular bodies, filaments, or reticula with variable sizes. Importantly, in VSV-infected cells, the nucleocapsid protein of the virus could blend with the MxA condensates in cells showing a concomitant antiviral phenotype [141]. Similarly, Mx1, the murine ortholog of human MxA, could also form nuclear condensates when being transfected into human cells. Interestingly, 20–30% of transfected cells also formed cytoplasmic giantin-based filaments, and these cells, but not the ones with only nuclear bodies, showed antiviral activity against VSV [142]. The mechanism underlying the antiviral effects of the cytoplasmic filaments formed by Mx1 remains unclear.

#### 2.4.4. Sequestration and Storage of Molecules

Cellular condensates work as compartments to selectively sequester biomolecules and stock them, which serves as an approach of resource conservation. For example, each proteasome consists of a catalytic core particle (CP) and a regulatory particle (RP). With yeast as a model, the proteasome holo-enzyme constituted by the CP and RP mostly stays in the nucleus in proliferative cells; however, in the quiescent state, they are transported into the cytoplasm and sequestered as protein condensates called proteasome storage granules (PSGs) [143]. The functions of PSGs include protecting yeast cells against stress and maintaining their fitness during aging [144]. When the cells exit quiescence, the PSGs will be disassembled and the proteasome will reenter the nucleus [145]. Furthermore, P bodies and stress granules are also able to sequester highly expressed mRNAs. Whether the stored mRNAs undergo translation or decay by individual cells in future can generate different phenotypes and improve their ability to withstand stress [146]. Meanwhile, P bodies and stress granules can also serve as protein quality control compartments that help cells to sequester misfolded proteins from the other cellular milieu [147].

Cellular condensates can also confiscate proteins to temporally curb their functions. For example, the death domain-associated protein (DAXX) is a chaperone of the histone H3.3 variant, and recruits HDACs to repress basal transcription [148]. Due to the interaction with PML, DAXX can be sequestered into the PML bodies to block its activity in repressing transcription, and sumoylation of PML is prerequisite for this process [148,149].

The nucleolus is a reputed storage apparatus in the nucleus and can sequester many regulatory proteins in response to different signals [150]. Many proteins involved in cell cycle progression, apoptosis and oncogenesis can be sequestered in nucleoli through different mechanisms. As a ubiquitination E3 ligase, MDM2 can be confined in nucleoli through its interaction with p14ARF or ATP molecules, which leads to p53 activation [151,152]. Another E3 ligase, VHL, can also be sequestered in the nucleolus in response to reduced extracellular pH. This can prevent the ubiquitination and degradation of its substrate HIF in the presence of oxygen, and allow it to activate its target genes [153]. Other important regulatory proteins with reported nucleolar sequestration include MYC, hTERT and CDC14 [154,155,156].

## 3. The Phase Separation of Proteins in Diseases

Accumulating evidence suggests that aberrant assembly of condensates is associated with cancers [157]. Below, we employ several examples to discuss how dysregulated phase separation of key regulatory proteins may contribute to neurodegenerative diseases and cancers.

### 3.1. LLPS and Neurodegenerative Diseases

#### 3.1.1. FUS

As a multifunctional DNA- and RNA-binding protein, FUS has been reportedly involved in transcription regulation, RNA splicing, RNA transport and DNA damage repair [158]. The FUS protein has an N terminal PrLD that is intrinsically disordered and critical to its phase-separated condensation [120,159]. The RNA-recognition motif (RRM) of FUS can bind to RNA molecules that promote FUS phase separation. Two domains are involved in FUS nuclear localization. First, the three RGG (arginine-glycine-glycine) repeats, designated as the RGG3 domain, can transport FUS from cytoplasm to nucleus. Second, the C-terminal proline tyrosine (PY) domain is a PY-NLS that can also promote FUS’s nuclear transportation, but it needs the assistance of the nuclear import receptor transportin, also known as karyopherin β2, to cross the nuclear pore complex [160,161]. Most FUS mutants showed impaired binding to the receptor transportin, leading to their increased cytoplasmic retention. The defective nuclear import of the FUS mutants causes their cytoplasmic aggregation in neuronal and sometimes glial cells, linked to disease pathogenesis, such as ALS [162].

ALS patient-derived mutations of G156E and R244C, located in or adjacent to the prion-like domain of the FUS protein, could convert its droplets to fibrous structures, which eventually form amyloid-like fibrillar aggregates and subsequently contribute to the protein misfolding diseases [31,163]. While the fusion of two adjacent wild-type FUS droplets could occur in seconds, the event would take many hours for the FUS(G156E) mutant [31]. Interestingly, the fibrillar aggregates of FUS(G156E) could act as seeds to efficiently induce the aggregation of wt FUS [163]. Both wt FUS and the G156E mutant could produce similar condensates in cells; however, in a rat model, FUS(G156E) mutant could create intranuclear inclusions in hippocampal neurons with cytotoxicity, likely due to the defects in regulating translation and RNA splicing [163].

It has been reported that the methylation of the arginine in front of the PY-NLS reduced FUS binding to the receptor transportin, and thus caused its cytoplasmic accumulation [164]. Interestingly, the arginine methylation of FUS also decreases its phase separation and stress granule association. Therefore, the NLS mutations of FUS in ALS patients not only weaken transportin-mediated nuclear import, but also abolish its arginine methylation, which promotes phase separation and stress granule formation of FUS [165]. Besides methylation, the PrLD of FUS can be phosphorylated by DNA-PK. The phosphorylated FUS protein exhibits reduced FUS phase separation and subsequently decreased aggregation tendency, which can ameliorate FUS-associated cytotoxicity [96].

#### 3.1.2. Tau

In 1975, Weingarten et al. isolated Tau as a protein essential for microtubule assembly [166], and the subsequent studies indicated this microtubule-associated protein as a regulator of axonal outgrowth and transport in neurons. Tau aggregation leads to the formation of intracellular fibrillary deposits that have been recognized as a hallmark of various neurodegenerative diseases, including Alzheimer’s disease, frontotemporal dementia and Parkinson disease, with a common name of tauopathies [167,168]. The intrinsically disordered property and phase-separation potential of Tau can be attributed to its high content of proline and glycine, and many polar and charged amino acids. The LLPS propensity of Tau is primarily controlled by the proline-rich domain in its middle region, which also contains many phosphorylation sites. Tau is a protein that harbors different posttranslational modifications, including phosphorylation, acetylation, glycosylation, glycation and ubiquitination [169]. Some of these modifications have been demonstrated to impact the LLPS of Tau through altering its net charge, conformation and interactions with other molecules. Hyperphosphorylation of Tau can promote the maturation of its condensates into insoluble amyloid-like fibrils contributing to the diseases [170]. Lysine residues are crucial for the LLPS of Tau, and thus their acetylation mediated by p300 and CBP can reduce its interaction with RNA and reverse its condensation [171]. Despite the repressive effects of acetylation on LLPS-mediated aggregation, acetylated Tau is associated with neurotoxicity because it shows dampened interaction with tubulin and impaired ability to promote the growth of microtubule filaments [172].

#### 3.1.3. TDP-43

TDP-43 was initially identified as a protein binding to a regulatory element in the long terminal repeat of HIV-1 and blockint the assembly of its transcription complex [173]. Other studies also revealed TDP-43 as an essential DNA/RNA-binding protein regulating RNA splicing [174]. Among ALS patients, 90–95% are sporadic, with mutations in the genes *C9ORF72*, *SOD1*, *FUS*, etc. Strikingly, about 97% of these ALS patients and 45% of FTLD patients exhibited TDP-43 aggregation, implicating its pathogenic role in causing the motor neuron diseases [175]. TDP-43 is one of the PrLD-containing proteins that are prone to aggregation. Either pre-mRNA alternative splicing or aberrant proteolytic cleavage of the full-length TDP-43 can generate the PrLD fragment, suggesting its high potential in forming aggregates [176,177].

Posttranslational modifications play a regulatory role in TDP-43 condensation. Despite the predominantly nuclear presence, TDP-43 phosphorylation is associated with its cytoplasmic translocation, which can drive early pathology of the diseases [178]. Hyper-phosphorylated TDP-43 tends to aggregate and generate inclusion bodies in the brains and spinal cords of the patients. Actually, phosphorylation of S409 and S410 has been considered a signature for ALS pathological analysis [179]. TDP-43 acetylation reduces its RNA-binding affinity and promotes accumulation of insoluble, hyper-phosphorylated TDP-43, which resembles the pathological inclusions observed in ALS and FTLD [180]. Additionally, ubiquitination of TDP-43 by its E3 ligase Parkin does not show clear degradation-orientated effects, but instead causes its cytoplasmic accumulation to form insoluble aggregates [181]. In addition, TDP-43 aggregation is associated with its C-terminal domain consisting of a prion-like glutamine/asparagine-rich domain and glycine-rich region that drives LLPS [175,182].

### 3.2. LLPS and Cancers

#### 3.2.1. SHP2

Src homology region 2 domain-containing phosphatase-2 (SHP2) is a non-receptor protein tyrosine phosphatase (PTP), encoded by the *PTPN11* gene. SHP2 contains two SH2 domains, a central PTP catalytic domain and a C-terminal tail. The two SH2 domains, C-SH2 and N-SH2, serve as phospho-tyrosine-binding regions to interact with the substrates [183]. As a ubiquitously expressed protein, SHP2 regulates many signaling pathways involved in mitogenic activation, metabolic control, and transcription regulation [184]. Germline mutations of SHP2 accounts for 50% of Noonan syndrome and 90% of LEOPARD syndrome (i.e., Noonan syndrome with multiple lentigines) [185,186] cases. Somatic SHP2 mutations are significantly associated with different human malignancies [187].

The intramolecular interaction between the N-SH2 and PTP domains serves as a “molecular switch” to block the phosphatase activity of SHP2. This switch can be turned on by the N-SH2 domain binding to specific phospho-tyrosine sequences of upstream growth factor receptors and/or scaffold proteins, leading to SHP2 activation. Mutations of SHP2 may either abolish the autoinhibitory switch or impair its PTP activity, which cause either Noonan syndrome or LEOPARD syndrome, respectively [188].

E76 is the most frequently mutated site of SHP2 in human cancers and the mutations disrupt the inhibition of PTP domain by the N-SH2, while R498 mutations in SHP2’s PTP domain are also commonly observed and associated with LEOPARD syndrome [189]. Interestingly, a recent study demonstrated that two disease-associated mutant proteins, SHP2(E76K) and SHP2(R498L), showed significantly increased tendency of droplet formation compared to the wild-type SHP2. Consistently, the two mutants also formed nuclear puncta in cells, but wild-type SHP2 did not [190]. However, unlike most previously reported proteins with LLPS capability, the SHP2 protein does not contain any IDR or repetitive multivalent modular domain. Interestingly, the catalytic PTP domain is also responsible for the phase separation of the SHP2 mutants. The mutations of N-SH2 enhance the PTP activity and subsequently promote ERK1/2 activation [190].

#### 3.2.2. YAP and TAZ

As downstream effectors of the Hippo signaling pathway, YAP (Yes-associated protein) and TAZ regulate many biological processes including cell proliferation, apoptosis and differentiation [191]. As transcription coactivators, unphosphorylated YAP/TAZ complexes can be translocated to the nucleus, and bind to the TEAD transcription factors that regulate the expression of several genes involved in cell proliferation and survival, such as MYC and BIRC5 [192]. In recent years, both YAP and TAZ have been demonstrated to undergo LLPS that plays an essential role in activating the expression of their target genes, subsequently promoting oncogenesis. The phase-separated condensates can help YAP and TAZ to compartmentalize transcription machinery, including BRD4, MED1, CDK9 and TEAD [193]. Noticeably, in the Hippo signaling pathway, LATS1/2 phosphorylate YAP at S172 and TAZ at S89 to increase their cytoplasmic retention, which can both inhibit the LLPS of YAP and TAZ and reduce their activity as coactivators [10,123,194]. It has been demonstrated that the Hippo pathway can be frequently inactivated through nonmutational mechanisms during oncogenesis [195], which may explain the consistent hyperactivation of the YAP and TAZ in various cancers [194].

## 4. Conclusions and Future Perspectives

In a cell, many different types of membraneless organelles or condensates existand provide relatively defined but still dynamic compartments for various biological reactions or material sequestration to occur in an undisturbed fashion. Biomolecules, biochemical reactions and various biological regulations are not present or happen in a chaotic or random manner in the complex cellular milieu. The concept of phase separation that regulates the formation of these compartments is likely a general mechanism to restrict biomolecules into particular compartments for designated biological activities. The questions concerning how large biomolecules, especially proteins and RNAs, are self-organized and undergo LLPS, and how their phase-separation capability can be linked and contribute to their specific activities, have intrigued many researchers and attracted increasing interest. Through the research endeavors over the past two decades, we have gained extensive knowledge regarding the molecular features, assembly requirements and material properties of these membraneless organelles or condensates. We have also obtained many insights in phase-separation-regulated biological processes, including biological reactions, resource storage or sequestration, and gene expression. Importantly, dysregulated LLPS of different proteins due to mutations or aberrant posttranslational modifications are causal causes of various human diseases, such as many neurodegenerative disorders and cancer. Despite the knowledge obtained from the reported studies related to LLPS in normal and diseased cellular conditions, many questions remain to be answered and fertile areas need to be explored. First, although IDRs are likely prerequisite elements for protein phase separation, there are still reported exceptions. Thus, how amino acid sequence and/or composition can precisely determine the LLPS properties of a protein needs to be further defined. Second, the sequences, secondary structures or other properties of nucleic acids involved in phase separation are still largely unexplored. Third, with neurodegenerative diseases as an example, the reasons accounting for the occurrence of phase-separated insoluble aggregates only observed in specific cell types deserve special investigation. Finally, we have only just begun to explore therapeutic applications that take advantage of the phase-separation mechanism for disease treatment.

## Figures and Tables

**Figure 1 ijms-23-05491-f001:**
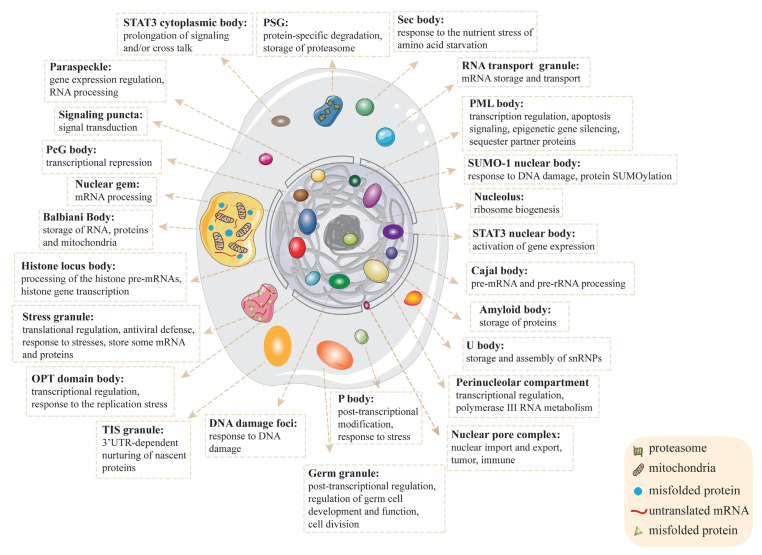
Schematic diagram of membraneless organelles and their functions in a eukaryotic cell.

**Figure 2 ijms-23-05491-f002:**
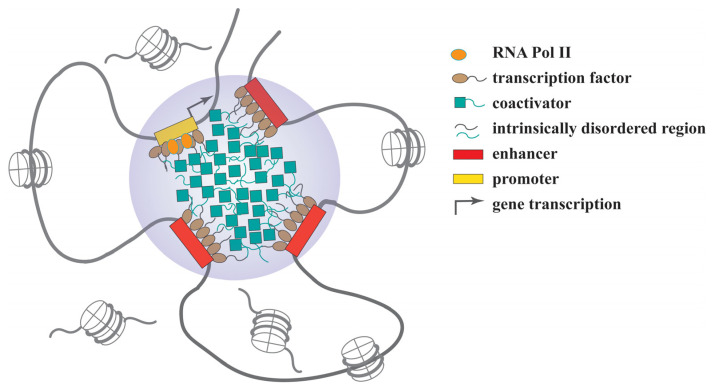
Regulation of gene expression by the phase-separation mechanism.

**Table 1 ijms-23-05491-t001:** Membraneless organelles and condensates assembled through the LLPS mechanism.

Localization	Name	Alias	Size (nm)	Components	Functions	Diseases	References
Cytoplasm	P-body	GW-body, RNA processing body, decapping body	100–300	K63, TRAF6, Tob1, TUT4, NoBody, LSM1, GW182, DDX3, DDX6, XRN1, etc.	mRNA degradation, post-transcriptional gene silencing, response to stress, storage of translationally repressed mRNAs	viral infection, neurodegenerative diseases, autoimmune diseases.	[39,40]
	Stress granule	—	1000–2000	RBPs, non-RBPs, TDRD3, TDP43, G3BP1, eIF3, eIF4G, PABPC1, etc.	translational regulation, response to stresses, antiviral defense, response to stresses, store mRNA and proteins	amyotrophic lateral sclerosis, frontotemporal lobar degeneration, cancer, viral infection, inflammatory diseases	[5,41]
	Germ granule	P-granule, chromatoid body, polar granule	250–4000	MEG-3, PGL, RNA, etc.	post-transcriptional regulation, regulation of Germ cell development and function, cell division	Germ cell development	[42]
	Synaptic density	Postsynaptic density	500	PSD-95, GKAP, Shank, Homer, etc.	responsible for signal processing	neuropsychiatric diseases	[43]
	RNA transport granule	Neuronal RNA granule	500–1000	Sam68, RNG105, SMN, etc.	mRNA storage and transport	neurodegenerative diseases	[44]
	Balbiani Body	Balbiani’s vesicle, the yolk body of Balbiani, yolk nucleus	50–250,000	RNA, mitochondria, Golgi, endoplasmic reticulum, etc.	store RNA, proteins and mitochondria	—	[45]
	Sec body	—	1000	COPII components, Sec16, etc.	response to the nutrient stress of amino acid starvation, protect ERES components from degradation	—	[46]
	U-body	Uridine-rich snRNP body	500	SnRNP, SMN, etc.	storage and assembly of snRNPs	spinal muscular atrophy	[47]
	PSG	—	500	proteasomes, free ubiquitin, etc.	protein-specific degradation, store proteasome	aging and age-related disease	[48]
	Signaling puncta	Dvl puncta	500–1000	Dvl-2, etc.	signal transduction	—	[49]
	Metabolic granule	G-body	1000–5000	glycolytic enzymes, etc.	glycolysis and storage	—	[50]
	STAT3 cytoplasmic body	STAT3 sequestering endosomes	—	STAT3	prolongation of signaling and/or cross talk	hepatoma	[51,52]
	TIS granule	—	1000–5000	TIS11B, membrane protein-encoding mRNAs	3′UTR-dependent nurturing of nascent proteins	—	[53]
Nuclear membrane	Nuclear pore complex	—	40–100	nucleoporins, NDC1, GP210, POM121 etc.	facilitate nucleocytoplasmic transport, chromatin organization	neurological disorders and the aging brain, viral infections and immunity, the development and progression of cancers	[54]
Nucleus	Nucleolus	—	1000–10,000	Nucleolin, rRNA, rDNA, etc.	ribosome biogenesis	Werner syndrome, Bloom syndrome, Treacher Collins syndrome, dyskeratosis congenita syndrome, Rothmund–Thomson syndrome	[55]
	HLB	—	1000	NPAT, FLASH, SLBP, p220^NPAT^, NELF, symplekin, etc.	processing of the histone pre-mRNAs, histone gene transcription	breast cancer	[56]
	DNA damage foci	—	500	γH2AX, ATM, 53BP1, RAD51, etc.	response to DNA damage	neurodegenerative diseases	[57]
	PML body	PML oncogenic domain, nuclear dot, Kremer body,	250–500	UBC9, RNF4, SP100, P53, DAXX, SUMO, PML, RNF168, etc.	transcription regulation, apoptosis signaling, epigenetic gene silencing, sequester partner proteins, SUMOylation sites	Acute Promyelocytic Leukemia, liver fibrosis	[58]
	Nuclear stress body	Peroxisome granule (PG)	300–3000	HSF1, HAP, SAM68, etc.	response to stress, control of gene expression and RNA splicing activities	metabolic syndrome	[59]
	Cajal body	accessory body	100–2000	RNA, snRNPs, scaRNAs, Coilin, SMN, etc.	pre-mRNA and pre-rRNA processing	amyotrophic lateral sclerosis, spinal muscular atrophy	[56]
	PcG body	—	200–1500	PRC1, PRC2, EZH2, etc.	transcriptional repression	malignant lymphomas, epithelial tumors	[60]
	CNB	—	1000–3000	CBP, SUMO-1, etc.	response to DNA damage, protein SUMOylation	—	[36]
	Paraspeckle	—	500–1000	CTN-RNA, PSP1, p54nrb, NEAT1, NONO, etc.	regulate gene expression, RNA processing	breast cancer, hepatocellular carcinoma, viral infection, neurodegenerative diseases	[61]
	PNC	—	250–4000	CUGBP, KSRP, polymerase III, Nucleolin, PTB, SRP RNA, etc.	transcriptional regulation, RNA metabolism	breast cancer, ovarian cancer	[62]
	Nuclear gem	Gemini of Cajal body, Gemini of coiled body	100–2000	SMN, etc.	mRNA processing	spinal muscular atrophy	[63]
	OPT domain body	53P1-OPT domain	1000–1500	Nascent mRNA, transcription factors, etc.	transcriptional regulation, response to the replication stress	—	[64]
	STAT3 nuclear body	—	—	STAT3, CREB binding protein (CBP), acetylated histone H4	activation of target genes	hepatoma	[51]
Nucleolus	Amyloid body	A-body	500–2000	Amyloid beta peptides, etc.	store proteins	neurodegenerative diseases	[65]

PSG: Proteasome storage granule; HLB: Histone locus body; PML: Promyelocytic Leukemia; PcG: Polycomb group; CNB: SUMO-1 nuclear body; PNC: Perinucleolar compartment; OPT: OCT1/PTF/transcription.

## Data Availability

The data presented in this study are available on request from the corresponding author.

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
