# Peer review of "Phase-Separated Subcellular Compartmentation and Related Human Diseases"

_ijms, 2022, doi:10.3390/ijms23105491_

Round 1
Reviewer 1 Report
Zhang and colleagues provide an extensive overview of phase-separated subcellular compartmentation and its role in human disease. The manuscript is well written and ready for publication.
Few minor points have to be addressed:
line 48,54, 102, 262, 278, 308, 362, 375, 408, 522, 554 references missed
line 77 formatting
Author Response
Reviewer 1:
Comments and Suggestions for Authors
Zhang and colleagues provide an extensive overview of phase-separated subcellular compartmentation and its role in human disease. The manuscript is well written and ready for publication.
Few minor points have to be addressed: line 48, 54, 102, 262, 278, 308, 362, 375, 408, 522, 554 references missed, line 77 formatting.
Reply: We thank the reviewer for the positive comments. We have added the missing references that the reviewer has pointed out. The formatting issue has also been corrected.

Reviewer 2 Report
Review of Zhang et al
This is a well-written overview of the rapid growing new area of research comprising LLPS and biomolecular condensates. The authors cover a great deal of the basic information in a way that is easy to understand and then expand it cover various known instances of membraneless organelles, their functions and involvement in a selection of disease states. Overall, the material covered is similar to that published in other reviews over the last 2 years by other authors.
Suggestions:
- While stress granules inhibit translation of mRNAs, there is a class of condensates which enhance mRNA translation. Please include a discussion of TIS11 granules.
- Many viruses now involve LLPS in their life cycle. Please provide a summary
- Please mention the family of IFN-induced antiviral GTPases MxA and Mx1 undergo LLPS in the cytoplasm and nucleus respectively, and that this determines their antiviral spectrum
- Please discuss that IL-6-activated Tyr-P-STAT3 forms nuclear bodies (reported in 2004), and new reports of PY-STAT3 cytoplasmic bodies plus validation that such structures represent condensates (in 2019). Note that PY-STAT3 is a very important transcription factor in immunology (activated by many cytokines) and also in cancer biology and thus “STAT3 nuclear bodies” cannot be left unmentioned.
- Please connect Pol II transcriptional regulation by CDK9 at nanodroplet with the recocognition that these occur at the Pol II RNA pause/termination sites originally identified using DRB. Look up Cisse’s papers.
Citations to items above and additional information can be found in Analyt Biochem (2020) 597, 113691
Author Response
Response to Reviewer 2Comments
Point 1: While stress granules inhibit translation of mRNAs, there is a class of condensates which enhance mRNA translation. Please include a discussion of TIS11 granules.
Reply: We thank the reviewer for the positive comments. We have added the description of the TIS granules generated by the TIS11B protein. The description is in the lines 364-384.
Point 2: Many viruses now involve LLPS in their life cycle. Please provide a summary.
Reply: We thank the reviewer for the suggestion. We have added a subsection of “2.4.3. Regulation of viral infection” to describe the functional interplay between viruses and host cells.
Point 3: Please mention the family of IFN-induced antiviral GTPases MxA and Mx1 undergo LLPS in the cytoplasm and nucleus respectively, and that this determines their antiviral spectrum.
Reply: We thank the reviewer for the suggestion. We have added the description of MxA and Mx1 for their antiviral activity. The description is in the lines 468-483.
Point 4: Please discuss that IL-6-activated Tyr-P-STAT3 forms nuclear bodies (reported in 2004), and new reports of PY-STAT3 cytoplasmic bodies plus validation that such structures represent condensates (in 2019). Note that PY-STAT3 is a very important transcription factor in immunology (activated by many cytokines) and also in cancer biology and thus “STAT3 nuclear bodies” cannot be left unmentioned.
Reply: We thank the reviewer for the suggestion. We have added the description of STAT3 nuclear bodies in the lines 427-438.
Point 5: Please connect Pol II transcriptional regulation by CDK9 at nanodroplet with the recognition that these occur at the Pol II RNA pause/termination sites originally identified using DRB. Look up Cisse’s papers. Citations to items above and additional information can be found in Analyt Biochem (2020) 597, 113691
Reply: As suggested by the reviewer, the description of CDK9 regulation in the context of Pol II-mediated transcription has been extended in the revised manuscript.
